# Collapse of the standard ferromagnetic domain structure in hybrid Co/Molecule bilayers

Mattia Benini [1,2] ✉, Andrei Shumilin [3,4] ✉, Viktor Kabanov [3], Rajib Kumar Rakshit[1], Antarjami Sahoo [1], Anita Halder[5,6], Andrea Droghetti[5,7], Francesco Cugini [8], Massimo Solzi [8], Diego Bisero [9], Patrizio Graziosi[1], Alberto Riminucci[1], Ilaria Bergenti [1], Manju Singh[1], Luca Gnoli [1], Samuele Sanna [10], Mirko Cinchetti [2], Tomaz Mertelj [3], Stefano Sanvito [5] ✉ & Valentin Alek Dediu[1] ✉

The interplay between Hund's coupling, exchange interaction and magnetic anisotropy is responsible for a multitude of magnetic phases, ranging from conventional ferromagnetism to exotic spin textures. Yet, engineering and fine-tuning a magnetic state remains a major challenge in modern magnetism. We show that the chemisorption of organic molecules over Co thin films offers a tool to transform the films from ferromagnetic to a glassy-type state. This emerges when the correlation length of the random anisotropy field, induced by the π-d molecule/metal hybridization, is comparable to the characteristic exchange length. Such a state is characterized by the collapse of the standard domain structure and the emergence of blurred pseudo-domains intertwined by diffuse and irregular domain walls. The magnetization reversal then involves topological vortex-like structures, which are here predicted and successfully measured by magnetic-force microscopy. At the macroscopic level this new glassy-type state is defined by a giant magnetic hardening and the violation of the magnetization-reversal Rayleigh law. Our work thus shows that the electronic interaction of a standard thin-film magnet with readily available molecules can generate structures with remarkable new magnetic properties, and thus opens a new avenue for the design of tailored-on-demand magnetic composites.

The microscopic understanding of the emergence of the macroscopic magnetic order is one of the earliest successes of quantum mechanics. In magnetism, Hund's coupling promotes the formation of the ions local magnetic moment, while the exchange interaction establishes how such moments align with respect to each other. The magnetic anisotropy couples the local moments with the lattice, selecting their preferential direction of alignment. These three ingredients alone give rise to a multitude of magnetic states with sharply different static and

[1]Istituto per lo Studio dei Materiali Nanostrutturati - CNR (ISMN-CNR), Via Piero Gobetti 101, Bologna, Italy. [2]Department of Physics, TU Dortmund University, Otto-Hahn-Straße 4, Dortmund, Germany. [3]Department of Complex Matter, Jozef Stefan Institute, Jamova 39, Ljubljana, Slovenia. [4]Instituto de Ciencia Molecular, Universitat de València, Paterna, Spain. [5]School of Physics, AMBER and CRANN Institute, Trinity College, Dublin, Ireland. [6]Department of Physics, SRM University-AP, Amaravati, Andhra Pradesh, India. [7]Department of Molecular Sciences and Nanosystems, Ca' Foscari University of Venice, via Torino, Venice-Mestre, Italy. [8]Dept. Mathematical, Physical and Computer Sciences, University of Parma, Parco Area delle Scienze 7/A, Parma, Italy. [9]Department of Physics and Earth Science, University of Ferrara, Via Saragat 1, Ferrara, Italy. [10]Department of Physics and Astronomy "A. Righi", University of Bologna, via Berti-Pichat 6/2, Bologna, Italy. ✉e-mail: mattia.benini@ismn.cnr.it; andrei.shumilin@ijs.si; SANVITOS@tcd.ie; valentin.dediu@cnr.it

dynamical properties, going from conventional ferromagnets, to various types of antiferromagnets, to more complex magnetic textures such as vortexes and skyrmions[1,2]. Crucially, in conventional magnetism, the exchange interaction and the magnetic anisotropy set the relevant length scales of a system and ultimately determine the dynamical response to an external perturbation.

It has been recently reported that the chemisorption of molecules on $3d$ metallic ferromagnetic layers can strongly modify the surface magnetic anisotropy and the exchange interaction, opening new routes for the manipulation of magnetic properties[3–6]. Such interfaces have been studied for years, with an almost-exclusive focus on the effects induced on the molecular layer[7–12]. Remarkably, at the opposite side of the interface the molecules promote specific surface orbital engineering, with their $p$-orbitals binding to selected $d$ metallic orbitals. This coupling induces radical modifications of the magnetocrystalline field in close proximity to the surface[13,14]. It has been shown that these hybridization effects can alter key magnetic properties of the $3d$ layer, resulting in the enhancement of the coercive fields for the in-plane magnetization rotation[5,15], the activation of new mechanisms for the in- to out-of-plane switching[13] and others[12].

Significant theoretical efforts have been devoted to understanding the interfacial effects in such systems and their possible propagation inside the ferromagnetic layer. Research has proceeded mainly via density functional theory (DFT) modeling of the effects induced by the proximity of a single molecule, with extrapolation to the entire surface[3,4,13–16]. This approach has successfully confirmed the enhancement of the local magnetic anisotropy[4,13,16] and the alteration of the exchange interactions[16] for the surface hybridized $3d$ atoms. Nevertheless, all theoretical reports have limited their predictions to purely local effects, excluding possible long-range correlation and any qualitative modification of the overall magnetism of the $3d$ ferromagnetic layer.

We show below both experimentally and theoretically that in hybrid Co/molecule thin film heterostructures a novel glassy magnetic phase is established, with properties radically different from those of the bare cobalt film. This phase emerges due to a correlated random-anisotropy field induced at the Co surface by the hybridization with the molecules. For in-plane magnetization the glassy magnetic structure displays blurred pseudo-domains intertwined by diffuse and irregular domain walls along with vortex-like topological defects perpendicular to the film plane. Intriguingly, the magnetization reversal in these systems is characterized by the violation of the Rayleigh law at low

magnetic fields, while at the opposite high field limit, it features a colossal enhancement of the magnetic anisotropy. This new finding opens conceptually new routes for the fabrication of novel materials with on-demand magnetic properties.

## Results

### Experiment: colossal enhancement of the coercive fields

We start by investigating the modifications induced to the in-plane magnetic properties of Co thin films ($d$ = 3, 5, and 7 nm) interfaced with two different molecular species, namely fullerene, $C_{60}$, and Tris(8-hydroxyquinolinato)Gallium, Gaq$_3$. The Co/molecule samples are compared to two reference systems, consisting of aluminum-protected Co/Al and oxidized by air exposure Co/CoO$_x$. While the first represents one of the best ways to investigate ex-situ bare-like cobalt films[17], the naturally oxidized Co samples allow us to compare the features of the Co/molecule composites with those of a fully inorganic interface. In what follows, we focus on 5 nm thick Co films, fully representative of all the claims advanced in this paper. Indeed, the behavior of 3 and 5 nm samples are very similar with respect to the accuracy of our experiments, while the results on 7 nm samples were already partly reported[5].

We start from the most common characterization of a ferromagnetic material, namely the hysteresis loops, which are here measured by MOKE from room temperature (RT) down to 80 K. While the detailed temperature dependence will be discussed later, Fig. 1 shows the results collected on the reference systems and on the two molecular-based cases at the intermediate and fully representative temperature of 150 K. The selected hysteresis loops of Fig. 1a clearly demonstrate the magnetic hardening induced by the molecular adsorption. The coercive fields, $H_C$, of Co/$C_{60}$ and Co/Gaq$_3$ are, respectively, 7-fold and 30-fold enhanced with respect to those of both the Al-capped and the CoO$_x$-interfaced Co thin films. The coercive-field enhancement produced in Co/$C_{60}$ is in qualitative and quantitative agreement with previously published data[14,15,18], indicating a strong hardening of the thin film. In contrast, the hybridization of the Co surface with Gaq$_3$ drives the hardening well beyond expectation, leading to a colossal enhancement of $H_C$. This persists even at room temperature, where the enhancement remains at about 100%. The coercive fields measured for 7 nm thick Co/molecule systems confirm the same trend but is characterized by a less pronounced hardening: at 80 K from 2- to 3-fold for Co/$C_{60}$ and Co/Gaq$_3$ respectively[5]. Noteworthily, while the hysteresis loop for Co/Gaq$_3$ in Fig. 1a shows some

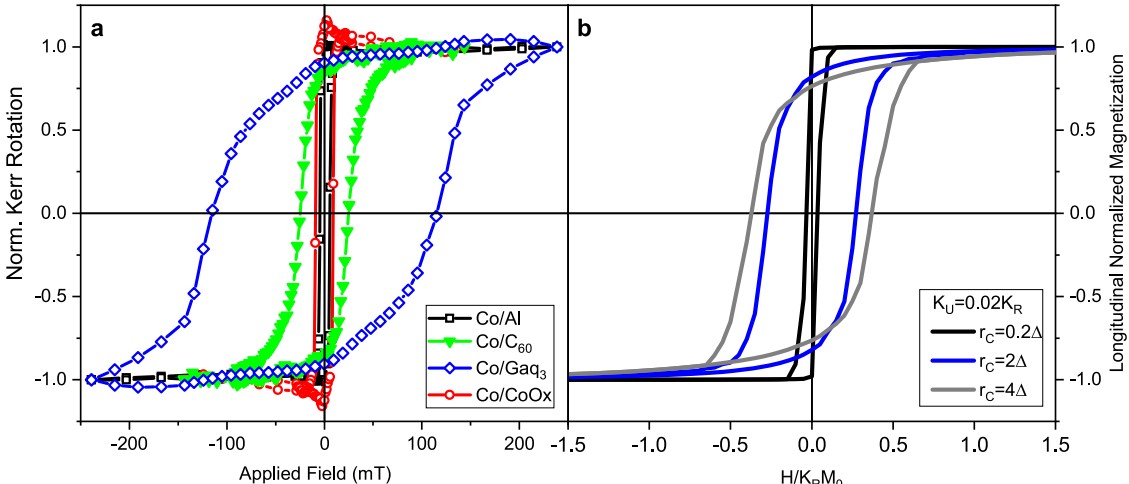

**Fig. 1 | Hysteresis loops. a** Experimental hysteresis curves for Co/$C_{60}$, Co/Gaq$_3$ and reference Co/Al, Co/CoOx systems measured at 150 K, showing the broadening of the loops for Co interfaced with molecular species. **b** Theoretical hysteresis curves

obtained with the correlated random anisotropy model for $K_R = 50K_U$ and different correlation radius values.

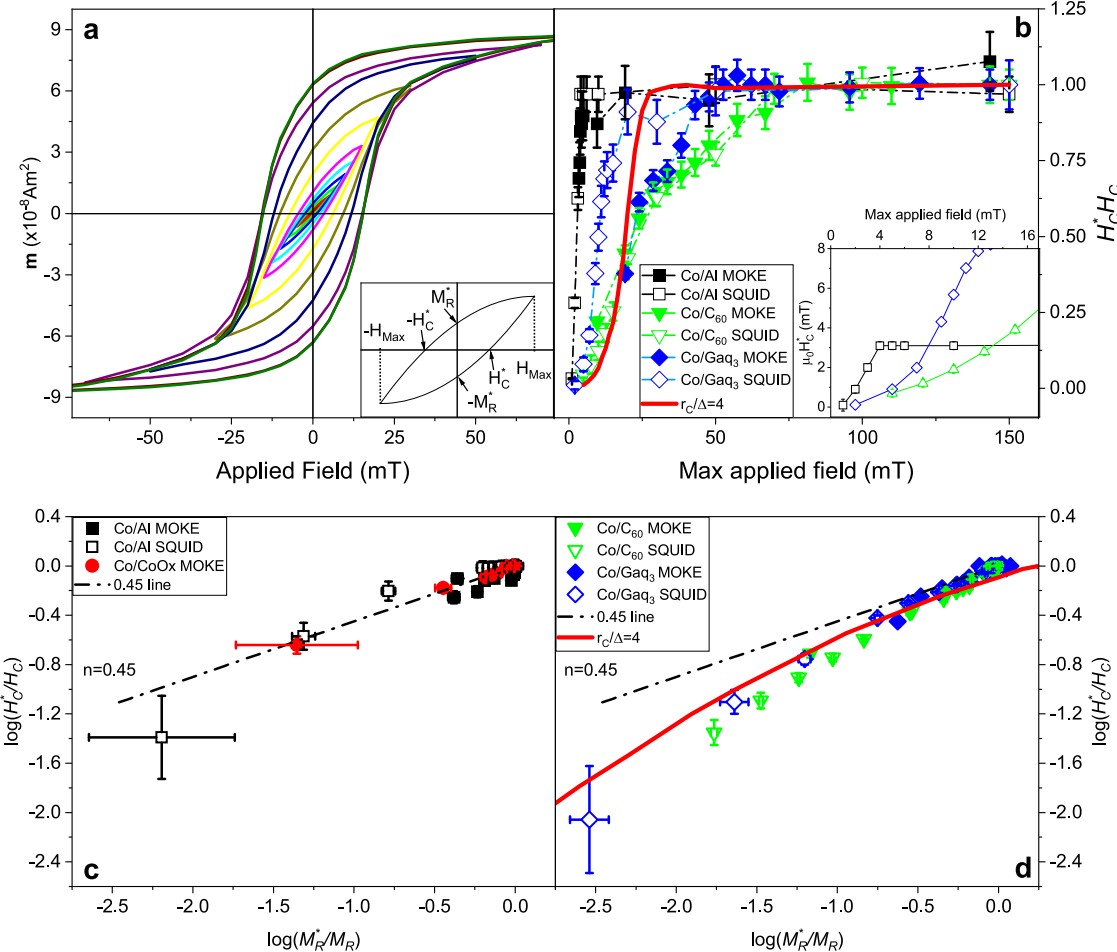

**Fig. 2 | Minor loops characterization. a** Example of minor loops obtained from SQUID magnetometry for a Co/C$_{60}$ system at T = 150 K; inset shows a sketch of a minor loop along with the quantities of interest for the numerical fit. **b** Minor loops coercive fields normalized by the saturated value for the systems investigated; continuous red line is calculated within the correlated anisotropy model with $r_C/\Delta = 4$. Inset shows the low-field trend of $H_C$ for representative systems. **c** log($H_C^*/H_C$) vs log($M_R^*/M_R$) trends for reference systems, well described by $n = 0.45$. **d** same plot for Co/Molecule systems showing a clear deviation from the standard behavior, while the correlated anisotropy model with $r_C/\Delta = 4$ provides a reasonably good fitting (red line). Error bars represent propagated uncertainty from single-measurement errors.

imperfections with respect to regular hysteresis shape, these deviations are due to MOKE-induced minor artifacts, which do not replicate in SQUID measurements (see below) and even in most MOKE loops detected on similar samples (see more details in the SI). This allows us to exclude any significant presence of a second (or more) magnetic phase, which could have introduced errors in comparing experimental data with theory.

The hysteresis loops presented in Fig. 1b are those calculated on the basis of the micromagnetic model that will be introduced later on. For the moment, note that the model accounts for both the colossal $H_C$ enhancement and the significant broadening of the magnetic reversal transition along the loops, two features that are clearly detectable experimentally for Co/C$_{60}$ and further magnified in Co/Gaq$_3$.

**Experiment: minor loops behavior**

Deeper insights into the observed magnetic hardening can be obtained from the hysteresis minor loops[19], as measured with MOKE (surface sensitive) and SQUID (bulk measurement) magnetometry. We start from a fully demagnetized state and proceed by subsequently increasing the field intervals of the magnetic loops, with the measurements taken at the representative temperature of 150 K for all the systems reported in Fig. 1a. While both techniques return very similar trends, SQUID measurements present more regular loops and allow us

to measure the absolute values of the magnetic moment. As an illustrative example, in Fig. 2a we show the SQUID raw data for the minor loops of Co/C$_{60}$ (with subtracted background). All the other MOKE and SQUID data can be found in the Supplementary Information (SI). The inset in Fig. 2a depicts a sketch of a typical minor loop and the parameters used for our quantitative analysis, namely the minor-loop coercive field, $H_C^*$, remanence $M_R^*$ and the maximum applied field $H_{MAX}$.

In Fig. 2b we report the relation between $H_C^*$ and $H_{MAX}$ for the reference and the molecule-interfaced cobalt layers, with data normalized to the average values of $H_C$ of the fully saturated loops (see SI for the non-normalized data). While the reference Co/Al samples are reproducibly characterized by a bisector-like growth at low fields, adsorbing molecules on Co leads to a radically different behavior. This is characterized by a non-linear and very gradual increase of $H_C^*$ towards the fully saturated loop value at $H_{MAX} \gg H_C$. The deviation between the reference and molecule-based samples is further emphasized in the inset of Fig. 2b, where a distinct transition from a linear (reference samples) to a power-law behavior (Co/molecule) indicates a drastic modification of the magnetization reversal mechanism at low magnetic fields.

We further analyse the minor loops data with the theoretical model of Takahashi et al.[20]. This is based on the well-known Rayleigh

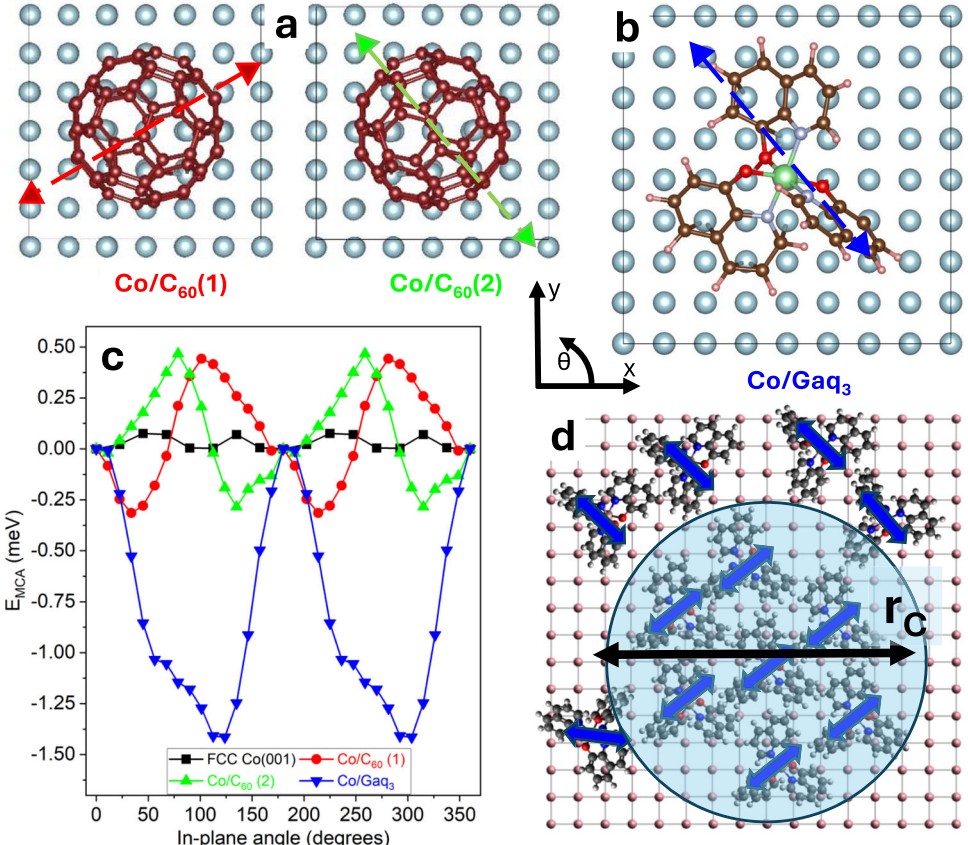

**Fig. 3 | Results of the DFT calculations. a** Top view of the two Co/C$_{60}$ adsorption geometries. **b** Top view of the Co/Gaq$_3$ slab, alongside with the frame of reference highlighting the easy axis obtained from the DFT calculation. The Co atoms are in gray. The atoms of the molecules are respectively represented by brown (C), red (O), cyan (N), pink (H), and green (Ga) spheres **(c)** E$_{MCA}$ as a function of the in-plane magnetization angle $\theta$ for two Co/C$_{60}$ and one Co/Gaq$_3$ slabs. The results are shifted so that $E(\theta = 0°) = 0$ meV. **d** A schematic representation of the correlation radius r$_C$. The arrows on the different figures indicate the local anisotropy axes.

relation[21–23] describing the magnetization reversal via domain wall (DW) motion,

$$M = \chi H + \eta H^2 \qquad (1)$$

where $\chi$ is the initial susceptibility and $\eta$ is the Rayleigh constant associated to the Barkhausen jumps. Starting from this equation, the model defines four scaling power laws with associated critical exponents, relating various quantities of the minor loops. Among those one can derive a relation between the coercive fields and the remanent magnetizations,

$$\frac{H_C^*}{H_C} = \left(\frac{M_R^*}{M_R}\right)^n \qquad (2)$$

where $n = 0.45$ is a universal exponent for all magnetic systems for which Eq. (1) is fulfilled. Such relation is typically satisfied by various ferromagnetic samples[24–26].

A log-log plot of the quantities entering Eq. (2) is provided in Fig. 2c, d for the reference and the Co/molecule samples, respectively. Note that data for both Co/Al and Co/CoO$_x$ are well compatible with the expected $n = 0.45$ exponent. In particular, the behavior of the oxidized sample is instructive, as it shows that the enhancement of the coercive field (in this case by surface oxidation) is not necessarily accompanied by a modification of Eq. (2) or more generally by any deviation from the Rayleigh law.

In contrast, when molecules are adsorbed on Co a significant deviation from the relation in Eq. (2) and hence from the Rayleigh law is observed. In fact, in Fig. 2d one can clearly see that the data for both molecular samples lie below the $n = 0.45$ line, and even certainly deviate from a power law. While deviations from the Rayleigh-derived laws can be observed at relatively high fields, where the Rayleigh law does not properly account for the magnetization dynamics[27], the Co/molecule systems are in disagreement with the expected trend for the whole field interval. This means that the magnetic behavior of the molecule-interfaced cobalt thin films is not well described by the conventional magnetization reversal mechanisms of ferromagnets. While the observed magnetic hardening can in principle be naively explained by a mere rescaling of the anisotropy constants, the deviation from Rayleigh law indicates the establishment of a new magnetic state displaying an anomalous magnetization reversal mechanism. We will now theoretically analyse these new magnetic properties by first looking at how the molecules modify the surface anisotropy, and then by investigating how such modification affects the reversal mechanism.

## Theoretical investigation: first-principles calculations

We start our theoretical analysis by performing first-principles DFT calculations for a supercell containing a four-layer Co slab with either a C$_{60}$ or a Gaq$_3$ molecule adsorbed on the top surface (see Fig. 3a, b). We explicitly focus on the in-plane magneto-crystalline anisotropy (MCA) and, by definition, our frame of reference has the z-axis perpendicular to the surface, which lies in the xy plane. We consider FCC rather than HCP Co based on the recent tunnel-electron-microscopy images of our samples[28] and for simplicity we take the [001] surface. The Gaq$_3$ molecule is adsorbed with two O and one N atoms pointing toward the Co surface (Fig. 3b). After a geometry optimization, the angle in between the two quinoline ligands closer to the surface opens up until

these ligands lay flat down on the surface, while the third ligand remains almost perpendicular to it[29,30]. For $C_{60}$ we consider two different adsorption geometries, labeled as 1 and 2, which differ by approximately a 45° molecular rotation (see Fig. 3a).

After the interface geometries are relaxed (see SI for more details), we estimate the MCA by rotating the Co magnetization in-plane and compute the system's energy $E(\theta)$ as a function of the magnetization angle, $\theta$, with respect to the x-axis. The direction of the minimum (maximum) energy corresponds to the slabs' easy (hard) axis, specified by the angle $\theta_{EASY}$ ($\theta_{HARD}$). The MCA energy is then defined as $E_{MCA} = E(\theta_{EASY}) - E(\theta_{HARD})$ and, by convention, the zero is set at the $\theta = 0°$ energy.

Note that the effect of the molecules on the MCA is purely electronic and it is induced by the hybridization of the molecular orbitals with the d shell of the surface cobalt atoms, strongly modifying the orbital populations in the latter. All the magnetic effects are calculated on the Co slab and can be decomposed in atomic contributions, highlighting the surface character[31] (see SI).

Fig. 3c shows the effects that the molecules' adsorption has on the MCA. When compared to the clean Co slab, which is almost magnetically isotropic ($E_{MCA} \approx 0.1$ meV), a marked uniaxiality is induced by the presence of the molecules, resulting in an extraordinary enhancement of the surface MCA energy, up to 0.7 meV for $C_{60}$ and 1.5 meV for Gaq$_3$. Moreover, not only does the anisotropy increase, but the orientation of the easy axis also becomes strongly dependent on the adsorption geometry of the molecule. In the case of Co/$C_{60}$, $\theta_{EASY}$ switches from 120° to 40° when the molecule is changed from configuration 1 to 2 (see Fig. 3c), while for Co/Gaq$_3$, this modification is further enhanced, since the ligands are strongly chemisorbed on the surface in a very asymmetric fashion, and the molecule assumes a rod-like conformation. These magnetic effects are found to be rather local, meaning that they stem from the surface atoms directly bonded to the molecule, whereas the surface regions and the Co atoms, which are not covered by the molecule, remain almost unaffected.

In summary, the DFT calculations explicitly show that the surface-molecule hybridization redefines the MCA. In particular, it introduces strong symmetry modifications by creating a local, at the molecular scale, in-plane uniaxiality. The MCA orientation then depends on the chemisorption geometry. Since multiple molecular conformations are possible on the surface, the net result of the depositing molecules is that of creating a random anisotropy field with a typical lengthscale comparable to the size of the molecules themselves.

## Theoretical investigation: macroscopic model

In order to describe the magnetization reversal dynamics we now move to micromagnetic model calculations. The described above local anisotropy encourages the application of the well-known empirical correlated random anisotropy model (RAM)[32,33]. The general concept of correlations in RAM models was used in a series of interesting works[34–36], but this approach was never employed for the description of such particular systems, consisting of fairly homogeneous magnetic thin films decorated by 2D randomly oriented islands of modified surface atoms. This expands the (calculated above) effect of the single molecule and distributes the local anisotropy over an area characterized by a certain correlation radius, $r_C$ (see Fig. 3d). The correlation radius $r_C$ is generated by molecular clusters with similar adsorption geometry and will be discussed in detail below.

Below we set the basis for the 2D Correlated Random Anisotropy Model (2D-CRAM), where the 2D limit indicates none or negligible variations of the magnetization along the axis perpendicular to the surface. This makes the model mostly adapted for the description of thin and ultra-thin magnetic films with random anisotropy.

We start from introducing the correlation term in the expression for the free energy:

$$\mathcal{F} = \frac{1}{2}\xi^2 \sum_\alpha (\nabla M_\alpha)^2 + K_Z(\mathbf{M} \cdot \mathbf{e}_z)^2 - K_R(\mathbf{M} \cdot \mathbf{a})^2 - K_U(\mathbf{M} \cdot \mathbf{e}_x)^2 - \mathbf{H} \cdot \mathbf{M}$$

(3)

Here $\mathbf{H}$ is the external magnetic field applied in the xy-plane, $\mathbf{M}$ represents the magnetization density and $M_\alpha$ is its projection to the cartesian axis $\alpha$. The parameter $K_Z$ encapsulates a simplified description of the MCA and acts to keep the magnetization in-plane. Typically, this is stronger than other anisotropy terms. In the simplest approximation $K_Z \to \infty$, hence $\mathbf{M}$ is allowed to rotate only in plane, keeping constant the absolute magnetization value $M_0$ and explicitly shaping our model for the 2D limit. The first term on the right-hand side of Eq. (3) accounts for the exchange interaction. It includes the magnetic exchange length, $\xi$, that is usually of the order of a few nanometers[37], reflecting the energy enhancement in non-uniform magnetization scenarios. The $K_R(\mathbf{Ma})^2$ term is central to our model, since it represents the random anisotropy averaged over the film thickness. While the absolute value of $K_R$ remains constant, the coordinate-dependent easy axis $\mathbf{a}(\mathbf{r})$ is random. The distribution of $\mathbf{a}(\mathbf{r})$ is dependent on the correlation length $r_C$, and its interplay with the renormalized magnetic length $\Delta = \xi/\sqrt{K_R}$ embodies the very core of the present model. The details on the correlation properties of $\mathbf{a}(\mathbf{r})$ with arbitrary $r_C$ are given in the SI. We also keep a small uniform anisotropy term $K_U$ defining the properties of the bare Co layer. This allows us to model Co samples without molecular overlayers by simply setting $K_R = 0$. We do not explicitly consider dipole-dipole interaction since it is suppressed in thin films with in-plane magnetization and we assume it to be irrelevant compared to the random anisotropy.

Considering the possibility to realize the long-scale correlation, we note that the growth of molecular layers on metallic or oxide surfaces typically involves tens of nm large nucleation islands[38,39]. The growth process is characterized by a finite diffusion time of the molecules on the surface, which allows them to bind in preferential, lowest energy configurations.

To find the magnetic properties of the 2D-CRAM, we perform numerical simulations based on the minimization of $\mathcal{F}$ and the Landau-Lifshitz-Gilbert equations. The analysis is performed considering only two key parameters: the dimensionless correlation length $r_C/\Delta$ and the ratio $K_U/K_R$, which describes the relation between uniform and random anisotropy. For details see the SI.

The first significant result is reported in Fig. 1b, which shows the dependency of the calculated hysteresis loops on $r_C$ at a fixed $K_U/K_R$ ratio (see SI for different $K_U/K_R$). For low correlation lengths the film properties are controlled by $K_U$, even when the random anisotropy is 50 times larger than the uniform one (see $K_u = 0.02K_R$, $r_C = 0.2\Delta$ curve in Fig. 1b). In this case the coercive field is low and the shape of hysteresis loop is almost square, similarly to what is observed experimentally for the reference Co/Al film in Fig. 1a. When $r_C$ is increased to values comparable to or larger than $\Delta$ the effects of the random anisotropy start to prevail: the hysteresis loops broaden, resulting in a strongly increased coercive field. Moreover, the shape also changes, displaying a broader, more gradual, magnetization reversal, similar to that of Co films interfaced with molecules in Fig. 1a. In the SI we show that, if the hysteresis loops for Co/$C_{60}$ and Co/Al samples are rescaled by setting the applied field in units of the closure field of Co/$C_{60}$, the data coincide with the model loops obtained for $K_U = 0.02K_R$ with $r_C/\Delta=4$ and $r_C/\Delta=0.2$, respectively.

In addition to the hysteresis loops, our model fairly accounts for the modification of the minor loops of the Co/molecule systems. It can be seen in Fig. 2 that the model adequately describes the minor loops

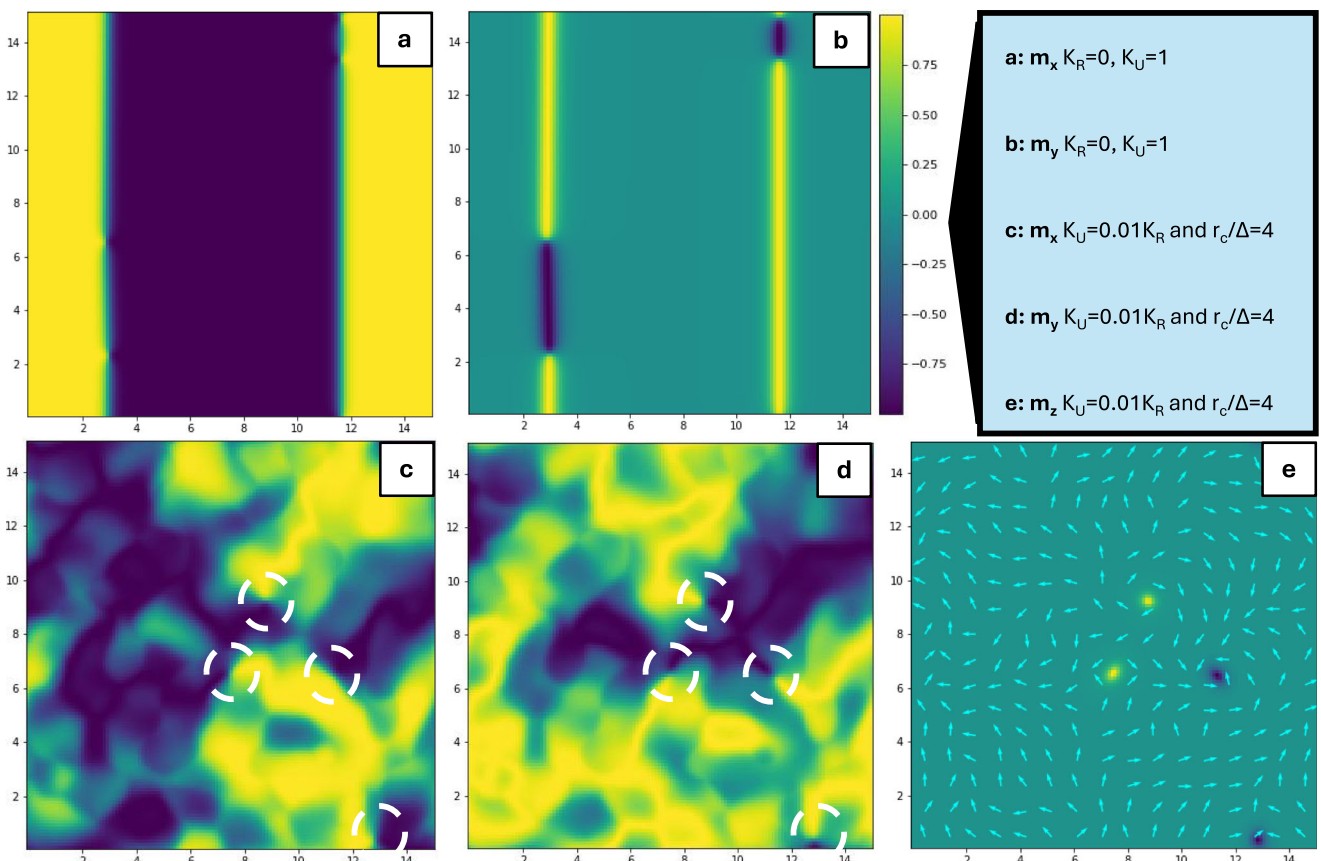

**a:** $m_x$ $K_R=0$, $K_U=1$

**b:** $m_y$ $K_R=0$, $K_U=1$

**c:** $m_x$ $K_U=0.01K_R$ and $r_c/\Delta=4$

**d:** $m_y$ $K_U=0.01K_R$ and $r_c/\Delta=4$

**e:** $m_z$ $K_U=0.01K_R$ and $r_c/\Delta=4$

**Fig. 4 | Micromagnetic model simulations.** Configuration of the normalized magnetization vector **m** components in the system's fully demagnetized state. Yellow (blue) indicates $m_\alpha = \pm 1$. **a** $m_x$ and (**b**) $m_y$ for the case $K_R = 0$ and $K_Z = 2K_U$ (i.e., no random anisotropy term): uniform anisotropy term dominates, and a standard domain configuration is achieved as expected. When the film properties are controlled by random anisotropy (**c**, **d**) the domain structure collapses and **m** continuously varies in random directions (depicted with white arrows). **e** For $K_R$ comparable to the shape anisotropy (in this case, $K_Z = 2K_R$) localized vortexes with finite z-component appear, their position being highlighted in (**c**, **d**) by white dashed circles.

evolution with field, capturing the non-linearity and the steady enhancement of the coercive field $H_C^*$ (Fig. 2b). Moreover, the application of the model describes very well the deviation from the power law in Fig. 2d.

Remarkably, our detailed micromagnetic simulations reveal an unusual glassy-type magnetic configuration for high correlation lengths. Even though in agreement with previous theoretical predictions[36,40] experimental evidences for such magnetic phase have never been reported before. Fig. 4 shows the maps of the normalized magnetization components $m_\alpha = M_\alpha/M_0$ with and without the random anisotropy term. The ferromagnetic film without a molecular overlayer is described by $K_R = 0$ and $K_U = 1$ (Fig. 4a, b - in this case $\Delta$ is defined as $\xi/\sqrt{K_U}$), and a standard magnetic configuration is clearly observed in its pseudo ground state, namely in the fully demagnetized state. For the magnetic field aligned along the x axis the division in domains oriented along the field (yellow) and opposite to it (blue) is well defined and the two are separated by narrow domain walls, with the presence of Bloch points. This magnetic configuration is standard for thin ferromagnetic films and the calculated domain wall width of 10-20 nm is in good agreement with the experimental data[37].

This domain configuration clearly collapses for finite $K_R$ and $r_C > \Delta$. For example, for $K_U = 0.02K_R$ and $r_C = 4\Delta$ a totally different magnetic configuration develops (Fig. 4c, d). It is characterized by a gradual transition between irregular islands of opposite magnetic orientations, separated by broad and diffuse transition regions (pseudo domain walls). The structures in Fig. 4c–e strongly depend on the cooling procedure, described in detail in the SI part.

Noteworthily, we confirm that the glassy state possesses also an additional and important characteristic fingerprint, i.e., the presence of topological defects[36,40], represented in our systems by magnetic vortexes with an out-of-plane orientation of the magnetization. This is found to happen at the singularity-like points in the x-y plane, where pseudo-domains with opposite orientations come in direct contact with each other (circles in Fig. 4d). The magnetization has a $2\pi$ rotation in the loop around this point, as shown in Fig. 4c–e. In addition, the predicted vortexes are evidenced by the z-component of the magnetization in Fig. 4e and in Fig. 5a–c. Along a hysteresis loop, the orientation of the vortexes magnetization depends on the magnetization history. For the virgin demagnetized state and during the initial magnetization both vortexes and anti-vortexes are present (Fig. 5a, d). However, considering that an inevitable small angle tilting of the magnetic field with respect to the film plane is present (Fig. 5d), during the field reversal for the positive branch of the loop all vortexes are oriented along the indicated perpendicular-to-plane component (Fig. 5b) and switch to the opposite direction at the reversed field (Fig. 5c).

Remarkably, high-resolution magnetic-force microscopy (MFM) images of the prototypical Co(5 nm)/$C_{60}$(25 nm) samples reveal a well-established system of vortexes at room temperature. While the detection of vortexes on the virgin magnetization curve has proven extremely challenging, we focused on the left and right hysteresis branches and performed the magnetic imaging at a small field of +2 mT applied after fully saturating the magnetization in the opposite direction. One can clearly see in Fig. 5e a number of vortex-like black magnetic defects, located in morphologically smooth areas (see the

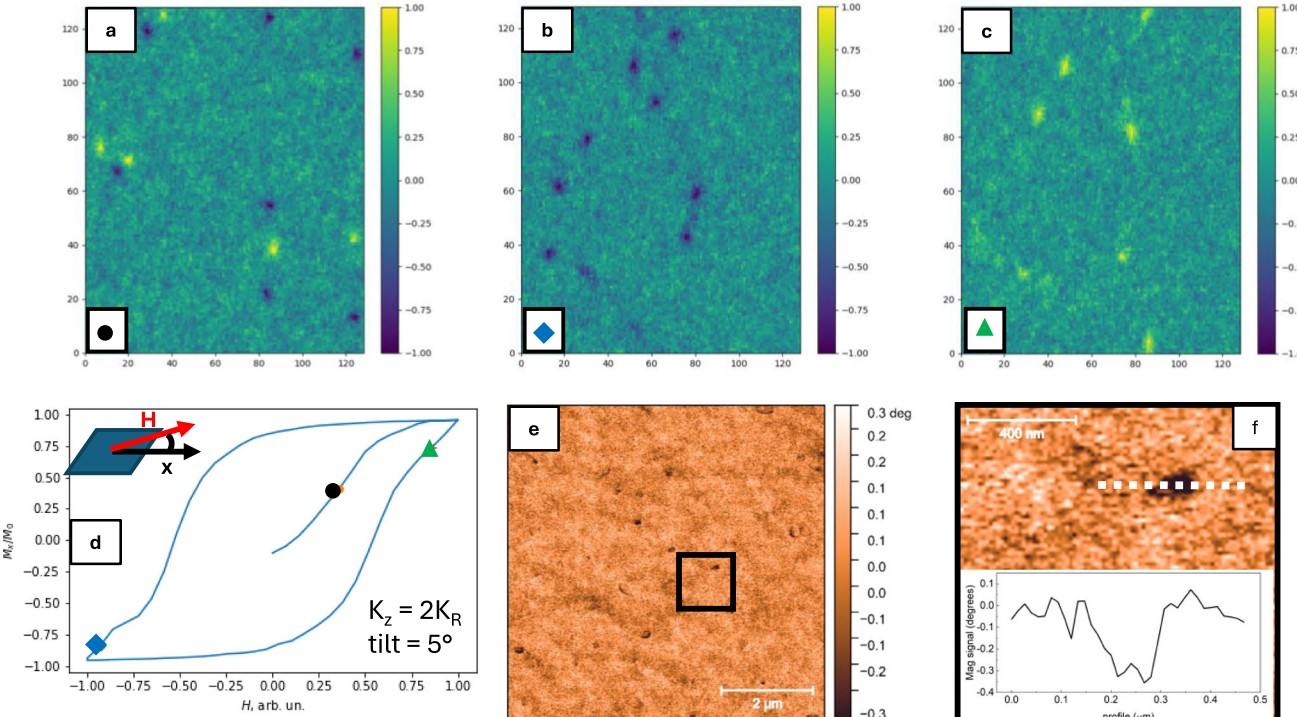

**Fig. 5 | Magnetic vortexes. a–c** Micromagnetic simulation of the spatial distribution of $m_z$ calculated at different representative points of a simulated hysteresis loop with an applied field tilted 5° out of plane (magnetization indicated in (**d**) by the circle, rhomb and triangle). Different vortexes configurations are predicted depending on the magnetization history. The MFM imaging of vortexes and their spatial distribution in the Co(5 nm)/$C_{60}$(25 nm) sample at RT and under an applied field of 2 mT is shown in **e**, where several vortexes (identified by the black spots) with lateral dimension of about 50 nm. The black square is shown zoomed in **f** for a better visualization of a representative dark spot, along with its lateral profile. The images in **e** and **f** were taken at the magnetization indicated by red star in (**d**).

details for all the observed vortexes and respective morphologies in SI). Zooming on a single vortex (Fig. 5f) reveals a typical size of 50-100 nm, an accuracy which cannot be exceeded in MFM studies. All vortexes are oriented in one direction (black color) in perfect agreement with magnetization history in Fig. 5d. For the sake of clarity, we show in SI the MFM magnetic imaging of the same system under the opposite field of −2mT, performed after fully saturating the system in the opposite direction. The contrast of the vortex-like defects turns from negative to positive (black to white), unambiguously demonstrating the magnetic origin and dependence of the observed magnetic signal on the magnetization history. The successful detection of vortexes by MFM imaging (Fig. 5e, f) represents a strong verification for the existence of this unusual magnetic phase and it will be discussed in more detail below. Note that the in-plane magnetization part cannot be detected by MFM, and it is generally extremely challenging considering the nanometer characteristic scales.

### Experiment: temperature dependence of the coercivity
Finally, we discuss the temperature dependence of the hysteresis loops obtained on several samples. Fig. 6 shows the extraordinary enhancement of the coercive fields when reducing the temperature from RT down to 80 K. This trend is particularly clear when compared to both reference samples, and it is in reasonable agreement with the Jiles-Atherton model[41], in line with previously reported trends measured on Co/$C_{60}$[15]. The exponential dependence of the coercivity with temperature is clearly visible in Fig. 6b and indicates no conceptual modification of the temperature behavior of coercivity with respect to the standard ferromagnetic state. Our data show no additional arguments either in favor or against the interesting hypothesis[18] of the significant modification of the metal-molecule hybridization at about 150 K due to the freezing of the rotational/vibrational molecular degrees of freedom. Nevertheless, the detection of vortexes at room

temperature allows us to claim that in Co/Molecule systems the glassy magnetic phase is present both below and above the possible freezing temperature. Note that the Co thin films interfaced with Gaq₃ proved to have less reproducible coercivities, differently from the Co/$C_{60}$ systems, which we tentatively attribute to the greater molecular complexity and hybridization geometry of Gallium-quinoline.

## Discussion
Currently, magnetism and spintronics have been significantly energized by the rise of numerous types of novel, and sometimes exotic, materials, such as topological insulators, 2D magnets, altermagnets etc. Yet, artificial compounds, where the conventional properties of their components are combined to display an anomalous behavior of the composite, represent a vast playground for magnetic materials with tailored-on-demand properties. We have here presented evidence that the hybridization of Co thin films with various molecules leads to the formation of a glassy magnetic phase, displaying a significantly enhanced magnetic anisotropy along with the violation of the Rayleigh relation at low fields, and other unusual properties.

Such a glassy state has been theoretically predicted time ago, but it has never been observed experimentally. For example, Chudnovsky and colleagues[34,40] anticipated a new magnetic phase, called Correlated Spin Glass (CSG) and characterized by multiple equilibria. They defined its order parameter and a number of uncommon properties, among which the lack of domain walls between Imry-Ma type domains[42], topological defects and others. In the previous paper[34] it is claimed that CSG has no magnetic remanence and no coercive field, while in the more recent paper it is shown that the magnetic order depends on the cooling-heating history and finite remance is likely to appear in realistic experimental conditions. Indeed, the zero-remanence requires a specific cooling of a demagnetized system[34] and its realization is experimentally very challenging. A similar model was

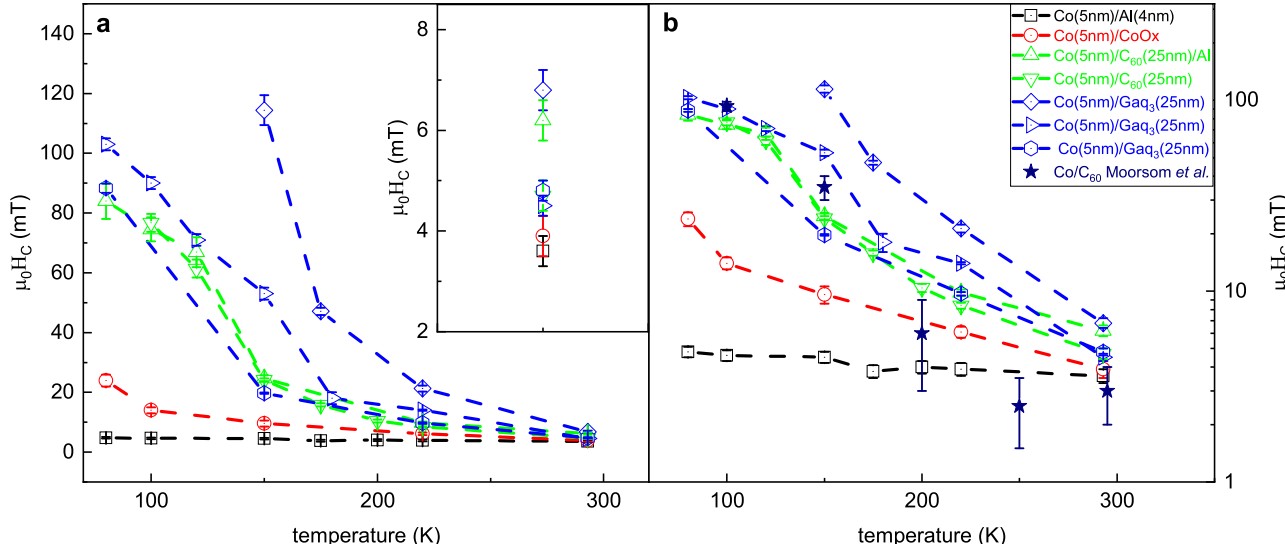

**Fig. 6 | Temperature dependence of the coercivity. a** The coercive field as a function of temperature for various $Co/C_{60}$ and $Co/Gaq_3$ systems, compared with reference Co/Al and Co/CoOx systems. Inset: enlargement of the values of $\mu_0 H_C$ at 293 K. **b** Same dataset reported in log-lin form, along with $Co/C_{60}$ SQUID data presented in Moorsom et al., [15], in reasonably good agreement with our experimental data. The authors thank Dr. Timothy Moorsom (University of Leeds) for sharing the data. Error bars indicate the resolution of the magnetic field sweep.

described by Dieny and colleagues[36] where the glassy shape of Imry-Ma domains is for the first time projected theoretically.

Our findings are in great agreement with these predictions and represent their first experimental realization. It is not surprising perhaps that this predicted and unobserved behavior was found by us in systems radically different from those to which RAM was routinely applied (amorphous[27,32,43] or nanocrystalline FM[44], magnetic alloys with impurities[45], ferromagnetic-antiferromagnetic bilayers[46] etc.). Moreover, our work shows that this glassy phase does not represent a purely academic interest, its properties are greatly appealing for various magnetic applications. The hardening effect presented in this paper and even stronger hardening communicated recently for similar systems[47] represent a novel and strongly versatile route for designing and building RE-free hard magnetic materials with on-demand parameters.

Scientifically, our research highlights the crucial importance of the correlation effects and respective length scales for the comprehensive understanding of the modifications of the magnetic properties in ferromagnetic thin films hybridized with molecules. Previous explanations of the $H_C$ enhancement were merely based on DFT calculations performed at the atomic/molecular scale. While able to justify qualitatively the magneto-crystalline enhancement, this approach could neither fit the hysteresis loops nor predict the violation of the Rayleigh law or other unusual macroscopic features.

Indeed, it is the formation of the glassy state which gives rise to this plethora of novel properties, from the hardening to the previously unreported violation of the Rayleigh law, from the topological defects to the collapse of the domain structure, etc. The previously proposed name CSG in our opinion is partly misleading and was perhaps inspired by the first calculations showing zero remanence. We believe that Correlated Ferromagnetic Glass (CFG) could be a more appropriate term to describe this phase (as also proposed in ref. [48]), but the correct terminology can be defined only in further debates and discussions.

Note that the CSG or CFG and all the related effects emerge only for $r_C/\Delta > 1$. Considering that the typical exchange length for thin cobalt films is in the range of 15-20 nm (our calculations and the FCC case in Vaz et al.[37]), and taking into account the definition $\Delta$ ($\Delta = \xi/\sqrt{K_R}$), for typical coercive field $H_C$ of 100 mT and dimensionless

$K_R = 1.4$, the ratio $r_C/\Delta > 1$ is obtained for $r_C > 10$-15 nm or equivalently 10-15 molecules bound in a correlated mode. The glassy phase is realized in the regime of intermediate thicknesses (approximately 3-10 nm), that is above the well-known in-plane to out-of-plane magnetization switching instability[13], and below thicknesses where the bulk properties strongly dominate the magnetization dynamics. Note that in a previous paper we demonstrated that the interface-induced magnetic hardening propagates to length scales of several nanometers[5].

We assign the microscopic origin of the glassy phase to the interplay between the anisotropy correlation length and the exchange length, both competing to define the shortest scale for the magnetization rotation and introducing an instability, which leads to the collapse of the standard domain structure. A similar interplay of competing lengths is fundamental in superconducting materials (competition between coherence length and penetration depth) or for metal-insulator phase transitions (competition between Bohr and screening radii).

In conclusion, our research demonstrates the emergence of the glassy ferromagnetic phase in a simple and easy to reproduce system. Its appearance is caused by a random but correlated anisotropy field established at the interface between metallic and molecular components for long-range correlations. The Correlated Spin Glass (or Correlated Ferromagnetic Glass, as proposed in this paper) features rich and complex new physics, calling for thorough further research, but it also represents a promising and versatile technology for the creation of conceptually innovative high-anisotropy magnetic materials.

## Methods
### Samples fabrication
Co/Molecule and Co/Al bilayers were grown in UHV conditions on $Al_2O_3(0001)$ single-crystal substrates. The Co deposition was done by electron beam evaporation (base pressure of $2.5 \times 10^{-9}$ mbar, rate 0.03 Å sec$^{-1}$) with the substrate kept at RT to promote the polycrystalline growth, confirmed in the previous publication[5]. The overlayers ($C_{60}$, $Gaq_3$, Al) were subsequently deposited, without breaking the vacuum, by thermal evaporation from effusion cells (base pressure of $1 \times 10^{-8}$ mbar) and the substrate kept at RT. Deposition rate: 0.15 Å sec$^{-1}$ for $C_{60}$, 0.25 Å sec$^{-1}$ for $Gaq_3$ and 0.1 Å sec$^{-1}$ for Al.

## Magneto-optical characterization

Hysteresis loops were recorded with an L-MOKE setup, the light wavelength $\lambda = 638.2$ nm (He-Ne laser). Samples were put in a cryostat and measured in a pressure of $10^{-6}$ mbar in vacuum condition. All temperature-dependent measurements were performed by cooling the systems in zero applied field. The hysteresis loops were subject to a symmetrisation procedure in order to remove quadratic Magneto-Optical signals. Hysteresis loops were taken with applied field varied in steps, with an effective field sweep rate of the order 1 mT/sec.

## SQUID characterization

The magnetic moment of Co/$C_{60}$, Co/$Gaq_3$ and Co/Al bilayers was measured using the DC option of a Quantum Design superconducting quantum interference device (SQUID) magnetometer MPMS-XL5. Hysteresis loops were recorded at 150 K with the magnetic field applied parallel to the film surface. The samples were first cooled with a zero applied magnetic field, and then a series of minor loops were recorded, increasing the maximum applied field.

## MFM characterization

In-field MFM images were recorded using the phase detection mode, i.e., monitoring the cantilever's phase of oscillation while the magnetic tip was scanning the sample surface at a distance of 50 nm (lift mode). In order to exclude the influence of the tip on the magnetic state of the sample, we used different scanning directions and tip to sample distances, obtaining the same results with different operating conditions.

## DFT calculations

The Co surfaces are modeled using periodically repeated slabs with a square surface supercell of $(4 \times 4)$ periodicity and four layers thickness. The molecules are adsorbed on one of the slab's surfaces. The calculations are spin polarized. The Perdew-Burke-Ernzerhof (PBE) generalized gradient approximation (GGA)[49] The geometry relaxations are carried using the Fritz Haber Institute ab-initio molecular simulations (FHI-aims) all-electron code[50] is selected for the exchange-correlation functional. A standard numerical atom-centered orbitals basis sets "tier 2" and a 3 x 3 x 1 **k**-point mesh is employed. The atomic positions of two bottom layers of the slabs are maintained fixed, while all other atoms are relaxed until the ionic force are smaller than 0.01 eV/Å.

After relaxation, the magnetic properties are predicted using the projector augmented wave (PAW) method[51] as implemented in the Vienna Ab-initio Simulation Package (VASP)[52] with the PBE exchange correlation functional. The tetrahedron integration method with a kinetic-energy cutoff of 600 eV is employed. An energy convergence criterion equal to $10^{-7}$ eV for total energy calculations is adopted. The **k**-point sampling is performed using a MonkhorstPack (MP) grid with $12 \times 12$ **k**-point mesh in the two-dimensional Brillouin zone. The energies $E(\theta)$ are obtained by means of the magnetic force theorem[53,54] followings two steps. Firstly, a scalar relativistic collinear charge self-consistent calculation is carried out to obtain the charge density. Then, that charge density is used as input in noncollinear calculations performed non-self-consistently including SOC, where the magnetization vector is oriented along different directions, and $E(\theta)$ is approximated as the band energy. The tetrahedron integration method with a kinetic-energy cutoff of 600 eV is employed. An energy convergence criterion equal to $10^{-7}$ eV for total energy calculations is adopted. The atomic SOC energies are defined as $E_{SOC} \sim \left\langle \frac{1}{r}\frac{dV}{dr}\mathbf{L} \cdot \mathbf{S} \right\rangle$ where $V(r)$ is the spherical part of the effective potential within the PAW sphere, and **L** and **S** are orbital and spin operators, respectively. These expectation values are about twice the actual values of the total energy correction to the second order in SOC[55].

## Data availability

The data used in this study are publicly available in the Zenodo database with the following link: https://doi.org/10.5281/zenodo.15525390.

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

## Acknowledgements

M.B., A.S., R.K.R., A.H., P.G., A.R., M.S., V.K., T.M., A.D., S.S. and V.A.D. acknowledge the support of the EC project INTERFAST (H2020-FET-OPEN-965046). L.G. and I.B. acknowledge the support of the EC project SINFONIA (H2020-FET-OPEN-964396). A. Halder was supported by the European Commission through the Marie Skłodowska-Curie individual fellowship VOLTEMAG-101065605. A.S. acknowledges the International Center for Theoretical Physics for supporting the ICTP-TRIL fellowship. A.S. acknowledges the financial support from the European Union (ERC-2021-StG-101042680 2D-SMARTiES). M.B and M.C. acknolwledge support of the Deutsche Forschungsgemeinschaft (DFG) through the project "Proximity", Project number: 556408835.

## Author contributions

M.B., S.S. (Stefano Sanvito) and V.A.D. mostly contributed to the drafting of the manuscript. M.B. and V.A.D. contributed to the coordination of all experimental work. R.K.R. and M.B. performed all sample preparation. M.B., I.B., and L.G. performed MOKE characterizations. F.C. and M.S. (Massimo Solzi) performed SQUID measurements. M.B. and P.G. performed data analysis of minor loops. D.B. performed MFM measurements. A.S. (Antarjami Sahoo), A.R. and M.S. (Manju Singh) performed magnetoresistance measurements. A.H., A.D. and S.S. (Stefano Sanvito) performed DFT modeling. A.S. (Andrei Shumilin), V.K. and T.M. developed the micromagnetic model. A.S. (Andrei Shumilin) and V.K. performed the micromagnetic simulations. M.C., S.S. (Samuele Sanna) and all other authors contributed to the interpretation of the experimental results.

## Funding

## Competing interests

The authors declare no competing interests.
