## [Peer review File · Nature Communications]

Collapse of the standard ferromagnetic domain structure in hybrid Co/Molecule bilayers

Corresponding Author: Dr Mattia Benini

Version 1:

Reviewer comments:

Reviewer #1

(Remarks to the Author)

Again, I appreciate Benini and co-authors revised their manuscript and provided a point-by-point response to the referees' comments. Here my comments focus on only two technical issues.

-Inconsistency between their experimental data.

See the data of Co/Gaq3 at 150K shown in "Fig. 1(a)" and Fig. S4.

I mentioned in the previous round, "This coercive field of 10 mT is saturated at the applied field of 100 mT and does not change even after applying 150 mT as shown in Fig. S4. No one can imagine that this value becomes equal to the coercive field more than 100 mT when the applied field is increased up to ~ 250 mT as the full loop of Fig. 1."

Namely, the Co/Gaq3 sample exhibited the coercive field of ~ 10 mT in Fig. S4. On the other hand, the same sample exhibited the coercive field more than 100 mT in Fig. 1(a). I asked about this remarkable issue last time, but unfortunately the author did not explain it.

Thus, I still question the quality of the data in this paper.

- About MFM observation.

The authors carefully explained how much difficult MFM observation is carried out. However, from the fact that the authors have successfully observed in a positive magnetic field, it is very clear that observation is not impossible.

Again, let me emphasize the following point. The authors want to insist that their experimental results provide the knowledge of remarkable new magnetic properties and opens a new avenue for the design of tailored-on-demand magnetic composites. To claim such findings, there must be no doubt about the experimental results. I do not change this opinion. The experimental evidence obtained by the MFM observation at a negative magnetic field, in which only the white contrast appear, is mandatory to support their conclusion.

In conclusion, I do not change my opinion and cannot recommend the publication.

Version 2:

Reviewer comments:

Reviewer #1

(Remarks to the Author)

I am satisfied with the authors' responses and the revisions. So, I am happy to recommend the publication in Nature Communications.

Collapse of the standard ferromagnetic domain structure in hybrid Co/Molecule bilayers

REPLY TO REVIEWER COMMENTS

Reviewer #1 (Remarks to the Author):

Again, I appreciate Benini and co-authors revised their manuscript and provided a point-by-point response to the referees' comments. Here my comments focus on only two technical issues.

1) -Inconsistency between their experimental data.

See the data of Co/Gaq3 at 150K shown in "Fig. 1(a)" and Fig. S4.

I mentioned in the previous round, "This coercive field of 10 mT is saturated at the applied field of 100 mT and does not change even after applying 150 mT as shown in Fig. S4. No one can imagine that this value becomes equal to the coercive field more than 100 mT when the applied field is increased up to ~ 250 mT as the full loop of Fig. 1."

Namely, the Co/Gaq3 sample exhibited the coercive field of ~ 10 mT in Fig. S4. On the other hand, the same sample exhibited the coercive field more than 100 mT in Fig. 1(a). I asked about this remarkable issue last time, but unfortunately the author did not explain it.

Thus, I still question the quality of the data in this paper.

Our Reply:

We apologize for the misunderstanding and believe that it was caused by our fault - the data on Co/Gaq3 in **Fig.1a** and **Fig.S4** were collected on two different samples.

Actually in our research we characterized tens of Co/Gaq₃ samples (not mentioning the other systems reported in this work), most of them having similar hardening effect. We hope this is reflected in the T dependence (Fig. 6) reported for 3 different Co/Gaq₃ samples. The minor loops characterization was done on two samples out of the pool we have. One was characterized by L-MOKE and the other with SQUID (**Fig. S2** and **Fig. S3**).

As also mentioned in previous responses, although the coercivity values differ, the shapes of the hysteresis loops and the trends shown in Fig. 2d are consistent with each other and, importantly, also with those observed in the Co/C₆₀ systems.

We did not hide, and actually the **Fig. 6** shows it, that while showing the highest hardening in some samples, the reproducibility of the Co/Gaq₃ systems was lower than that of Co/C₆₀ systems. We added in this last revised version an explicit comment on this in the discussion of **Fig.6**.

Moreover, our recent paper (ref 15) has explicitly demonstrated the Co/Mol systems do not split in bulk and interface components. The paper showed that even for thicker, 7 nm films, the magnetic anisotropy is changed homogeneously for the whole sample, as no double phase behaviour is observed by L-MOKE, and the ^{59}Co NMR peaks are shifted along the power coordinate and not broadened. It is clear to us that the interface cobalt atoms are strongly modified by molecules and this may/shall constitute ultrathin interfacial inclusions with modified anisotropy or exchange, but such inclusions cannot be detected in simple and apparently not in static experiments.

2) - About MFM observation.

The authors carefully explained how much difficult MFM observation is carried out. However, from the fact that the authors have successfully observed in a positive magnetic field, it is very clear that observation is not impossible.

Again, let me emphasize the following point. The authors want to insist that their experimental results provide the knowledge of remarkable new magnetic properties and opens a new avenue for the design of tailored-on-demand magnetic composites. To claim such findings, there must be no doubt about the experimental results. I do not change this opinion. The experimental evidence obtained by the MFM observation at a negative magnetic field, in which only the white contrast appear, is mandatory to support their conclusion.

Our Reply:

We must thank the Reviewer for insisting on this - we succeeded to detect exactly what requested and agree that now our claims are better supported! The MFM imaging at the opposite field of the same intensity (-2 mT), after fully saturating the system in the opposite direction, revealed exactly white contrast vortices of similar density, size and magnetic profile. We added a comment on this in the main text (line 325, above **Fig. 5**) and on the SI, where the MFM image and the vortices profiles are added (**Fig. S12**). We really hope this fully answers the last technical concern and hope to have convinced the reviewer on the robustness of our claims.

Summarising, we would like to acknowledge the scientifically rigorous and perfectly correct comments from the reviewer, helping us to rise the manuscript to the necessary level.